# Nanoscale imaging of super-high-frequency microelectromechanical resonators with femtometer sensitivity

Daehun Lee [1], Shahin Jahanbani [1], Jack Kramer [2], Ruochen Lu [2]✉ & Keji Lai [1]✉

Implementing microelectromechanical system (MEMS) resonators calls for detailed microscopic understanding of the devices, such as energy dissipation channels, spurious modes, and imperfections from microfabrication. Here, we report the nanoscale imaging of a freestanding super-high-frequency (3 – 30 GHz) lateral overtone bulk acoustic resonator with unprecedented spatial resolution and displacement sensitivity. Using transmission-mode microwave impedance microscopy, we have visualized mode profiles of individual overtones and analyzed higher-order transverse spurious modes and anchor loss. The integrated TMIM signals are in good agreement with the stored mechanical energy in the resonator. Quantitative analysis with finite-element modeling shows that the noise floor is equivalent to an in-plane displacement of 10 fm/√Hz at room temperatures, which can be further improved under cryogenic environments. Our work contributes to the design and characterization of MEMS resonators with better performance for telecommunication, sensing, and quantum information science applications.

Microelectromechanical system (MEMS) resonators convert electrical energy into the mechanical domain through piezoelectric transducers and store it as mechanical vibration[1]. Due to the micrometer-sized wavelength and small acoustic loss of gigahertz (GHz) elastic waves, these chip-scale structures exhibit very high quality ($Q$) factors[1–3]. In the super high frequency (SHF, 3–30 GHz) regime, high-$Q$ piezoelectric thin-film resonators are utilized as a frequency reference for consumer electronics and scientific instruments[2], building blocks for high-rejection radio frequency (RF) filters in wireless telecommunication[3], and quantum information elements for the generation, transfer, and entanglement of quantum states[4–6]. In particular, acoustic resonators featuring a series of densely packed resonances[7] are highly sought after for multi-frequency systems such as comb filters[8,9], reconfigurable oscillators[10], and multi-phonon quantum acoustic sources[4,6]. Successful implementation of such resonators calls for a microscopic understanding of the devices, such as mechanical mode profiles, energy dissipation channels, spurious modes, and imperfections from microfabrication.

Unfortunately, such features are mostly overlooked by conventional transducer-based electrical readouts, which becomes the major bottleneck for designing efficient acoustic or cross-domain microsystems.

In order to meet this pressing demand, it is desirable, albeit rather challenging, to directly visualize the acoustic mode profiles at SHF. First, since the SHF acoustic wavelength is on the order of 1 μm, a spatial resolution better than 100 nm is necessary for local imaging. Secondly, the vibration amplitude drops at high operation frequencies due to increased stiffness for structures with reduced dimensions. As a result, a sensitivity floor well below 1 pm is required for sensitive equipment with low excitation signals[2] or quantum acoustic systems with low phonon numbers[4,6]. Finally, many acoustic modes in prevailing piezoelectrics are primarily associated with in-plane oscillations. Techniques that measure out-of-plane displacements, regardless of the sensitivity level, are not applicable under such circumstances.

To date, a number of microscopy tools have been utilized to probe GHz acoustic fields with notable limitations. For instance,

[1]Department of Physics, University of Texas at Austin, Austin, TX 78712, USA. [2]Department of Electrical and Computer Engineering, University of Texas at Austin, Austin, TX 78712, USA. ✉e-mail: ruochen@utexas.edu; kejilai@physics.utexas.edu

acoustic imaging by scanning electron microscopy (SEM) is limited to sub-GHz frequencies due to the charging effect[11]. Methods using atomic-force microscopy (AFM) rely on nonlinear effects to detect surface displacements, which does not favor studies with low signal levels[12,13]. Stroboscopic X-ray imaging requires highly coherent X-ray sources, restricting the use and limiting the sensitivity[14,15]. The incumbent acoustic imaging tools are based on laser technology, such as homodyne[16–18] and heterodyne[19–21] interferometry and pump-probe techniques[22–24]. These optical methods share a diffraction-limited spatial resolution of ~ 0.5 μm and only detect vertical displacement. In addition, for transparent piezoelectric membranes in SHF resonators, the optical contrast is usually too weak to enable highly sensitive measurements. A new method capable of imaging laterally polarized SHF acoustic waves with high sensitivity and spatial resolution is thus crucial to advance our knowledge of MEMS resonators and implement devices with better performance.

In this work, we report the visualization of acoustic waves in a freestanding LiNbO₃ thin-film resonator with equally spaced tones around 5 GHz using transmission-mode microwave impedance microscopy (TMIM)[25–27]. Specifically, we use a lateral overtone bulk acoustic resonator (LOBAR)[9,28] as the testbed, where transducer electrodes only cover a small section of the acoustic cavity to excite the overtone response. LOBARs differ from conventional resonators in their low acoustic and thermoelastic damping due to the small footprint of electrodes, and, thus, the higher $Q$-factors for each tone[3]. The mapping of mode profiles provides direct information on individual tones with a spatial resolution on the order of 100 nm, from which the acoustic wavelength and phase velocity are quantitatively extracted. Higher-order spurious modes and resonator anchor leakage are observed and analyzed. The integrated TMIM signals are in good agreement with the stored mechanical energy inside the LOBAR device. A comparison between the TMIM and finite-element simulation results indicates that the equivalent sensitivity of in-plane displacement can reach an unprecedented level of 10 fm/√Hz. Our work paves the road to optimize the design of MEMS resonators and explore quantum acoustic devices.

## Results

The LOBAR device in this work is fabricated on a transferred 510 nm thick 128° Y-cut LiNbO₃ membrane released from a silicon wafer[29].

First-order antisymmetric (A1) Lamb mode[30], where the thickness-shear vibration is dominantly in-plane, is excited by the lateral electrical field between electrodes via the large transverse piezoelectric coefficient $d_{15}$ of 128° Y-cut LiNbO₃[31]. Unlike fundamental modes such as shear horizontal (SH0)[32,33] that suffer from sub-200 nm feature size at SHF, the operating frequency of A1 is collectively determined by the lateral dimension and film thickness. Consequently, A1 features fast phase velocity, large $k^2$, and low acoustic damping, ideal for frequency scaling into the SHF regime[30,31]. Fig. 1a shows the optical image of the fabricated device, where the 50 nm-thick aluminum interdigitated transducers (IDTs) are deposited on top of the thin film. The admittance response measured by a vector network analyzer (VNA) is plotted in Fig. 1b. More than 20 resonances between 4.4 and 5.6 GHz are efficiently excited, each corresponding to a lateral overtone of the LiNbO₃ membrane cavity. The inset of Fig. 1b shows both the measured and simulated admittance using frequency-domain finite-element analysis (FEA)[29]. The excellent agreement between the VNA data and simulated curves highlights the effectiveness of FEA modeling. The mode index is plotted in Fig. 1c, showing a linear dependence on the resonant frequency. Small discrepancies presumably result from the slight mode distortion caused by the IDTs and device imperfections (Supplementary Note 1). The results demonstrate that the LiNbO₃ A1 LOBAR is an excellent platform for multi-frequency applications in the SHF regime.

Spatial imaging of acoustic waves on this LOBAR device is carried out in our atomic-force microscopy (AFM) based TMIM setup[25–27], as schematically shown in Fig. 2a. The TMIM can cover a broad frequency range of 0.1–18 GHz. The A1 wave is excited by the IDT and reflected by the free boundaries on the sides, forming standing waves at corresponding resonant frequencies. The TMIM tip behaves as a microwave receiver to pick up the GHz piezoelectric potential[25]. Note that the tip can be modeled as an electrically floating metal sphere with ~100 nm diameter. Its perturbation to the surface potential is thus negligible. The signal is then amplified and demodulated by an in-phase/quadrature (I/Q) mixer using the same microwave source. If we express the RF and LO inputs to the mixer as $V_{RF} \propto e^{i(\omega t - kx)}$ and $V_{LO} \propto e^{i(\omega t + \phi)}$ ($\omega$: angular frequency, $k$: acoustic wave vector, $\phi$: mixer phase), respectively, the two TMIM output channels are $V_{Ch1} \propto \mathrm{Re}(V_{RF}V_{LO}^*) = \cos(kx + \phi)$ and $V_{Ch2} \propto \mathrm{Im}(V_{RF}V_{LO}^*) = -\sin(kx + \phi)$. In other words, the complex-valued signal $V_{Ch1} + i^*V_{Ch2} \propto e^{-i(kx+\phi)}$ pro-

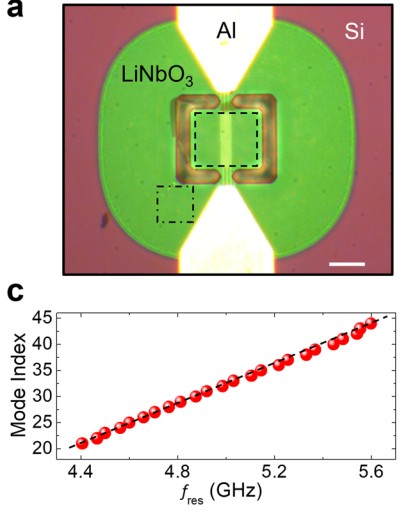

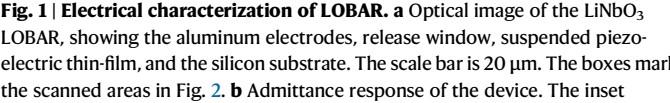

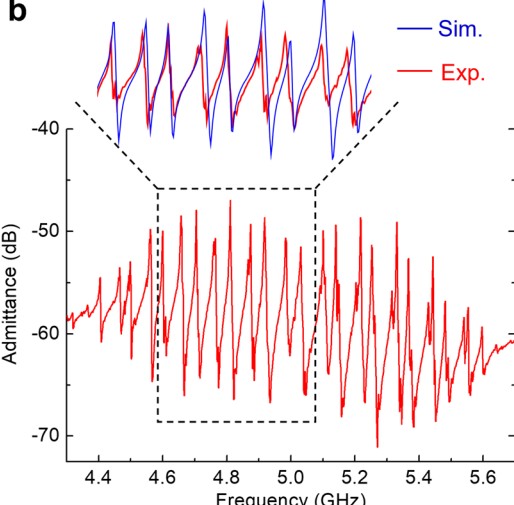

**Fig. 1 | Electrical characterization of LOBAR. a** Optical image of the LiNbO₃ LOBAR, showing the aluminum electrodes, release window, suspended piezoelectric thin-film, and the silicon substrate. The scale bar is 20 μm. The boxes mark the scanned areas in Fig. 2. **b** Admittance response of the device. The inset compares the experimentally measured data and FEA simulated results within 4.6–5.1 GHz. **c** Mode index as a function of resonant frequency for the overtones in (**b**). The dashed line is a linear fit to the data.

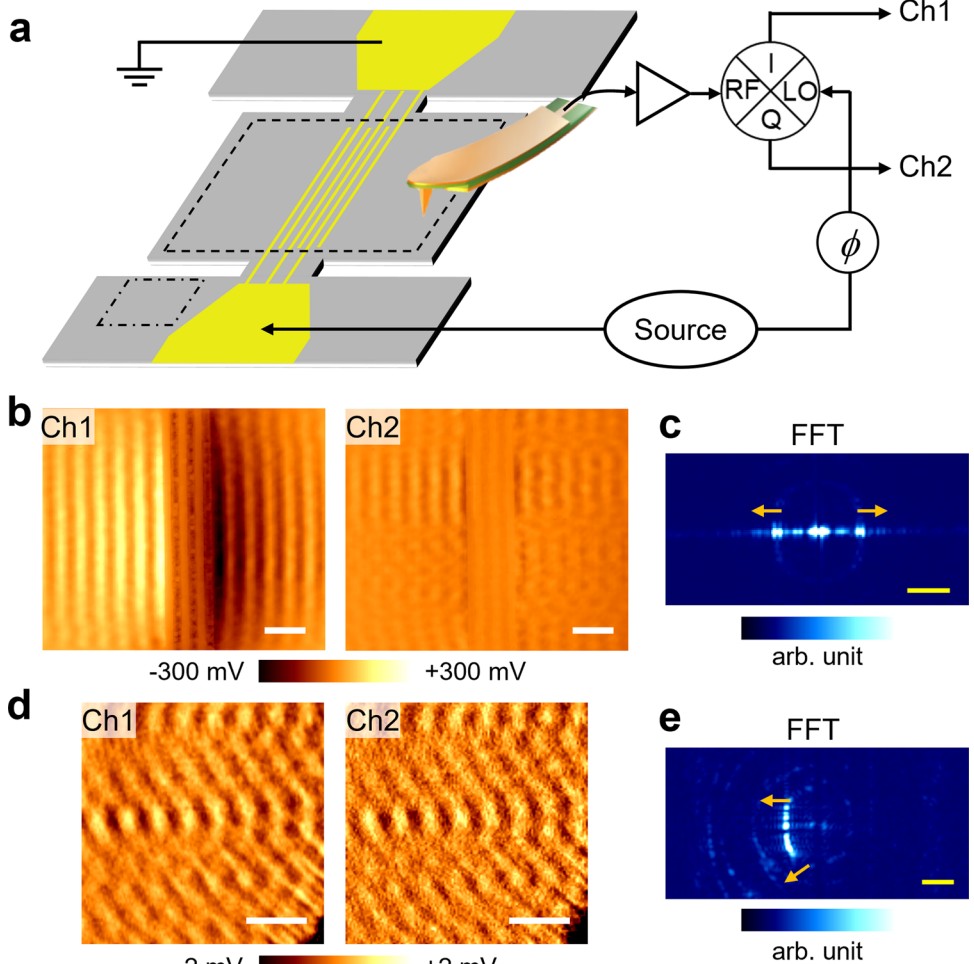

**Fig. 2 | TMIM imaging of the LOBAR device and FFT analysis. a** Schematics of the TMIM setup and LOBAR device. **b** TMIM images inside the resonant cavity, marked by the dashed box in (**a**). **c** Corresponding FFT image of (**b**). **d** TMIM images near a corner of the device anchor, marked by the dash-dotted box in (**a**). **e** Corresponding FFT image of (**d**). The arrows in (**c**) and (**e**) indicate the wave propagation direction in $k$-space maps. Scale bars in (**b**) and (**d**) are 5 μm. Scale bars in (**c**) and (**e**) are 0.4 μm$^{-1}$.

vides a phase-sensitive measurement of the demodulated electric potential on the sample surface, proportional to the piezoelectric coefficient and local displacement field. Moreover, fast Fourier transformation (FFT) of the real-space TMIM images will yield information in the reciprocal space ($k$-space)[33,34], which is crucial for analyzing wave behaviors.

The TMIM images at $f = 4.978$ GHz (the 32nd overtone) on the LOBAR device are displayed in Fig. 2b. To avoid sample damage, we scan an area of 35 μm × 30 μm that is slightly smaller than the size of the suspended membrane (38 μm × 32 μm). The scan speed is 3.33 s per line (half time in the forward scan and the other half in the backward scan). The contact force for AFM feedback is kept below 1 nN to minimize mechanical load on the sample. The TMIM results vividly demonstrate the excitation of A1 Lamb waves on LiNbO₃, whose signals are much stronger than that on the IDT electrodes. The FFT image in Fig. 2c indicates that the leftward and rightward traveling waves are equal in magnitude, i.e., the net displacement is a standing wave inside the resonator. For comparison, Fig. 2d shows the TMIM images near one corner of the IDT anchors. The corresponding FFT image in Fig. 2e suggests that the acoustic wave here propagates away from the resonant cavity. Such acoustic leakage through the anchor is a key parameter for MEMS resonators[35,36]. From our data, the TMIM signal strength at the anchor is two orders of magnitude lower than that inside the resonator, corresponding to a

loss of acoustic power of ~ 10⁻⁴. Given that the $Q$-factor is around 1000 for this mode, we conclude that the anchor loss is not the main source of energy dissipation in our device. The fact that TMIM can readily resolve the small leakage signifies the exquisite sensitivity of our technique, which will be quantified later. In addition, the TMIM data in the unsuspended region of the device indicates that the spatial resolution is indeed on the order of 100 nm (Supplementary Note 2).

In order to obtain spatially resolved information on various overtones, we have performed broadband TMIM imaging on the LOBAR device. Note that for standing waves, we can adjust the mixer phase $\phi$ such that most signals are in one TMIM channel[25], which will be presented hereafter (Supplementary Note 3). Figure 3a, b shows the AFM and TMIM images at 5 selected overtones, respectively. As the mode index increases, the spacing between adjacent fringes is decreased. Using the quasi-static approximation, one can show that $f \approx \sqrt{(v_l/\lambda)^2 + (v_t/2t)^2}$ ($\lambda$: acoustic wavelength, $t$: film thickness, $v_l$ ~7 km/s: longitudinal group velocity, $v_t$ ~ 4.2 km/s: thickness-shear group velocity)[29], which matches well with the TMIM data in Fig. 3c. Furthermore, the phase velocity $v_{ph} = \lambda \cdot f \approx \sqrt{v_l^2 + (v_t\lambda/2t)^2}$ decreases as decreasing $\lambda$ or increasing $f$, consistent with the results in Fig. 3d. We emphasize that the high $v_{ph}$ (~ 12 km/s at 5 GHz) of the A1 mode makes

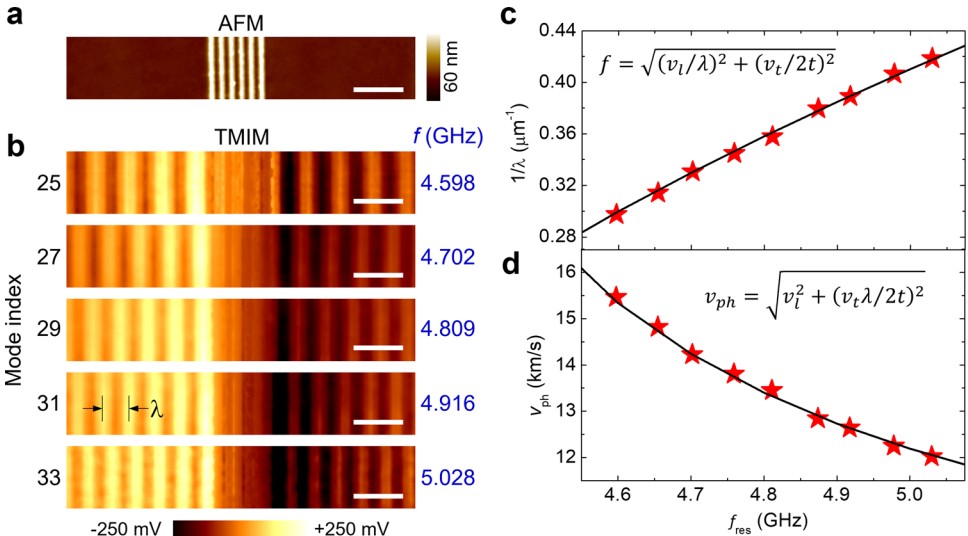

**Fig. 3 | Broadband imaging of multiple overtones. a** AFM image at the center of the LOBAR device, showing the IDT fingers. **b** TMIM images at five selected overtones. The corresponding frequencies are listed on the right. All scale bars are 5 μm. **c** Measured $1/\lambda$ and **d** phase velocity as a function of the resonant frequency. The lines fit the equations based on quasi-static approximations.

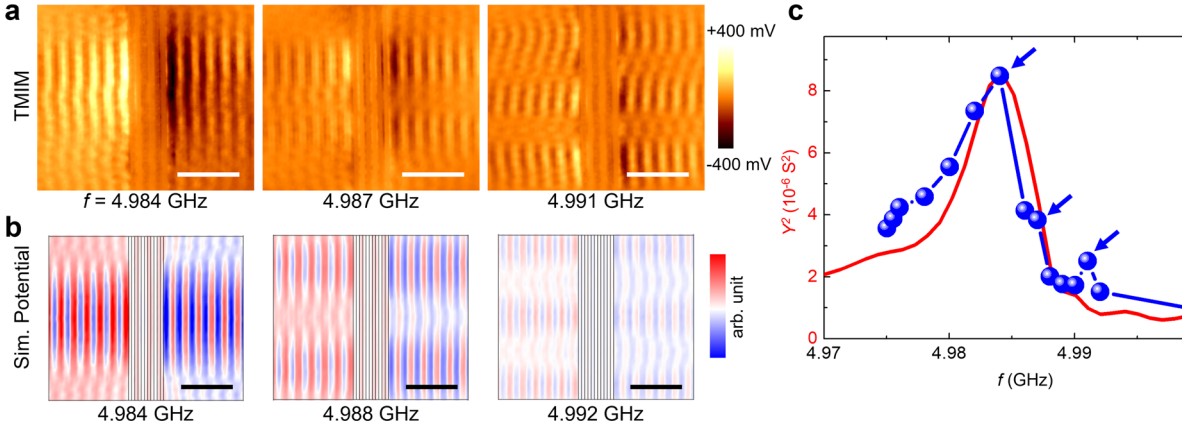

**Fig. 4 | Higher-order transverse modes and *Q*-factor. a** Left to right: TMIM images corresponding to the fundamental, 2nd, and 3rd transverse modes of the 32nd overtone of the LOBAR device. **b** Simulated potential maps that match the experimental data. The slight difference in frequencies between (**a**) and (**b**) is likely due to uncertainties in device parameters. Note that the scanned area in **a** is 35 μm × 30 μm, whereas the simulated area of the full membrane is 38 μm × 32 μm. All scale bars are 10 μm. **c** TMIM-Mod² signals integrated over the scanned area. The square of admittance measured by VNA is plotted for comparison. The positions of the main tone and higher-order modes are indicated by blue arrows.

the IDT feature size much easier for microfabrication than that of the SH0 mode ($v_{ph}$ ~ 4 km/s)[32,33]. The TMIM imaging experiment directly visualizes such crucial information.

We now turn to the evolution of acoustic profiles near one particular resonance of the LOBAR device. Figure 4a shows three TMIM images near the 32nd overtone. The formation of transverse standing waves inside the resonant cavity is clearly seen at frequencies above the main tone[37]. The FEA simulated acoustic patterns of the fundamental, 2nd-order, and 3rd-order transverse modes are displayed in Fig. 4b, which agrees with our measurement. Since higher-order transverse modes usually reduce the *Q*-factor of the fundamental mode, they should be avoided, or their frequencies should be pushed away from the main tone in the resonator design[37]. Note that these

undesired spurious resonances are different from the ones due to the nonlinear dynamic response of the drive mode[38,39]. In addition, given that the TMIM signal is proportional to the displacement field, the square of the TMIM modulus (Mod² = Ch1² + Ch2²) provides a good measure of the local acoustic power. Figure 4c plots the TMIM-Mod² signals integrated over the entire scanned area as a function of frequency. The result agrees with the measured $Y^2$ (square of admittance), which is also proportional to the acoustic power inside the resonator. A *Q*-factor of ~700 can be estimated from either the integrated TMIM-Mod² signals or the $Y^2$ data. Interestingly, the TMIM-Mod² data also indicate the existence of higher-order modes as two small peaks above the main tone, as marked by the blue arrows. Visualizing these spurious modes and frequency-dependent study of their behaviors are invaluable for optimizing future LOBAR structures.

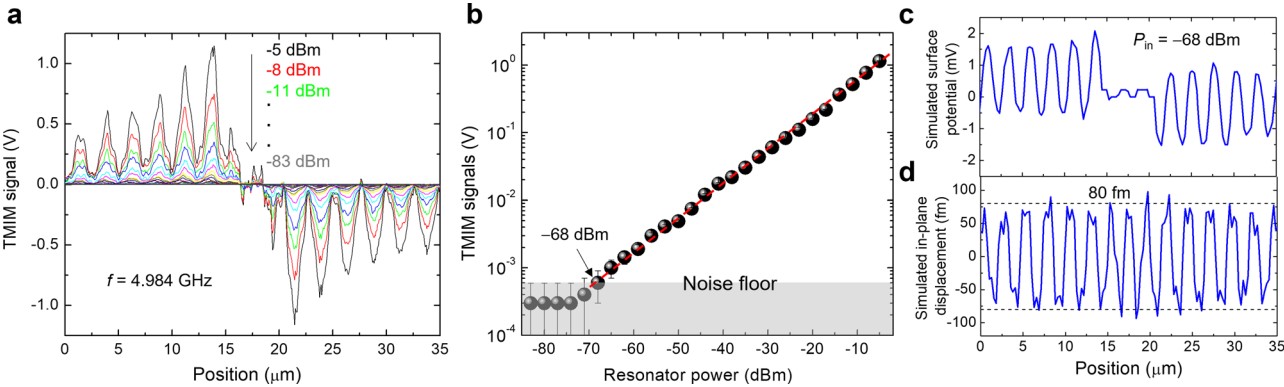

**Fig. 5 | Analysis of TMIM sensitivity. a** TMIM line profiles across the center of LOBAR at various input powers. **b** Maximum TMIM signal in (**a**) as a function of input power to the resonator. The noise floor corresponds to $P_{in} = -68$ dBm. The error bars represent the standard deviation. **c** Simulated surface potential and **d** in-plane displacement across the center of LOBAR at the same input power.

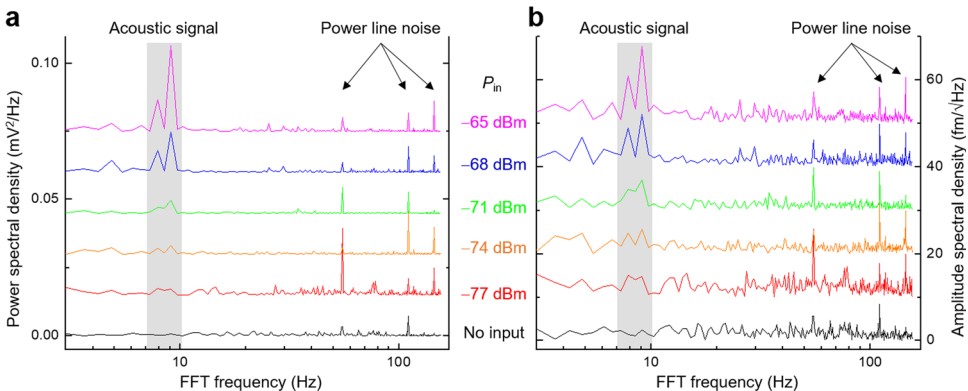

**Fig. 6 | Noise power analysis of TMIM data. a** Power spectral density (PSD) of selected TMIM line scans with different input power $P_{in}$. **b** Amplitude spectral density (ASD) of the same data in (**a**). Peaks associated with the acoustic signals in this LOBAR device and power-line noise are labeled in both plots.

The excellent agreement between TMIM and FEA results allows us to evaluate the ultimate sensitivity of our instrument in terms of displacement fields. To this end, we include an impedance-match section[25] to route the tip impedance to 50 Ω (Supplementary Note 4) such that the sensitivity is further enhanced at $f$ ~5 GHz. Figure 5a shows the TMIM line profiles across the center of the LOBAR at various input power $P_{in}$ delivered to the resonator. In Fig. 5b, we plot the maximum TMIM signal (first peak near the IDT) as a function of $P_{in}$, showing that the instrument noise floor is reached at $P_{in} = -68$ dBm. A few TMIM line-scan data around this input power can be found in Supplementary Note 5. The GHz voltage on the IDT electrode is given by $\sqrt{P_{in} * Z_0}(1 + \Gamma)$, where $Z_0 = 50$ Ω and $\Gamma$ is the reflection coefficient measured by VNA. We then use FEA to simulate the piezoelectric potential (Fig. 5c) at the same IDT voltage, which indeed resembles the measured line profile. Moreover, the simulated in-plane displacement field in Fig. 5d suggests that the corresponding oscillation amplitude at this input power is ~80 fm. Finally, by comparing the peak TMIM signals and the simulated amplitude of in-plane displacements at all input powers, we obtain the conversion factor of ~ 0.01 mV/fm in our current setup, as detailed in Supplementary Note 5.

To quantitatively evaluate the sensitivity of our instrument, we have carried out the standard noise power analysis[40] of the TMIM data. Figure 6a shows the power spectral density (PSD) curves of selected TMIM line scans. The acoustic signals in this LOBAR device are easily identified as peaks at 7–10 Hz in the Fourier transformation spectrum since they are oscillatory in space and thus oscillatory in time during line scans. The PSD peaks at ~ 55 Hz, and higher harmonics correspond to the power-line noise. Using the conversion factor above and taking

the square root of PSD, we obtain the amplitude spectral density (APD) curves in Fig. 6b, from which a number of key features can be identified. First, the noise floor of our current setup is limited by the power-line noise of 10 fm/√Hz, which is already superior to the best value obtained by optical methods by more than a factor of 5[21,24]. Secondly, the APD of the acoustic signal at $P_{in} = -68$ dBm is just above that of the power-line noise, consistent with the plot in Fig. 5b. Finally, the level of broadband noise in Fig. 6b is 3 ~ 5 fm/√Hz, which sets the ultimate sensitivity floor of our instrument. In other words, the acoustic signals cannot be observed for $P_{in}$ below −74 ~ −77 dBm even after filtering out the power-line noise. Further improvement of the sensitivity requires the lowering of such Johnson-Nyquist white noise[41,42]. Since the thermal noise amplitude scales with the square root of temperature, the ASD of white noise will drop from 3 to 5 fm/√Hz at room temperature (300 K) to below 1 fm/√Hz under liquid-helium temperatures (~ 4 K) and below 0.1 fm/√Hz at dilution-refrigerator temperatures (<100 mK). Once the TMIM is implemented under cryogenic environments[43,44], the technique with unprecedented sensitivity will find widespread applications in the burgeoning field of quantum acoustics.

In summary, we demonstrate the nanoscale acoustic imaging of a freestanding overtone resonator with femtometer sensitivity by microwave microscopy. By mapping the piezoelectric surface potential in both real-space and reciprocal-space images, we obtain direct information on the acoustic profile of individual tones, anchor leakage, and spurious transverse modes. The stored acoustic energy can be evaluated by integrating the TMIM-Mod² signals inside the resonator. Our quantitative analysis shows that the TMIM can detect 5 GHz in-plane oscillations in $LiNbO_3$ down to a level of 10 fm/√Hz. Further

optimization of the technique and implementation under cryogenic temperatures are expected to reach the quantum limit of single phonon detection. Our work represents a major advancement in acoustic imaging in terms of spatial resolution, displacement sensitivity, and operation frequency, which are important for communications, sensing, and quantum information applications.

## Methods

### Device fabrication
The 510 nm 128° Y-cut LiNbO$_3$ thin film, which was thinned down from a single-crystal wafer and transferred onto a 4-inch silicon (Si) wafer, was provided by NGK Insulators, Ltd.[29]. Such films retain high crystallinity and bulk material properties[45]. The release window was defined by electron beam lithography (EBL) and plasma etching. The top electrodes were fabricated by EBL and lift-off of 50-nm-thick aluminum. Finally, the LiNbO$_3$ thin film was released from the Si substrate by a XeF$_2$ etcher.

### Finite-element modeling
We simulated the in-plane displacement and potential profiles by finite-element methods using COMSOL Multiphysics 6.0 software, which is widely used for the simulation of acoustic thin-film resonators. The solid mechanics module and electrostatic module are coupled in the Piezoelectric Effect Multiphysics module, allowing us to compute the mechanical and electrical response. The software solves elastic wave equations in a linear piezoelectric medium with strain-charge coupling. Material properties in the simulation are provided by the COMSOL Material Library.

### Experimental setup
The transmission-mode microwave impedance microscopy (TMIM) is implemented on an atomic-force microscopy platform (ParkAFM, XE-70). The shielded cantilever probe (Model 5–300 N) is commercially available from PrimeNano Inc. Details of the TMIM experiments can be found in ref. [25].

## Data availability
All data supporting the findings of this study are available within the article and/or the SI Appendix. The raw data are available from the corresponding author upon reasonable request.

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

## Acknowledgements
The TMIM work was supported by the NSF Division of Materials Research Award DMR-2004536 and the Electrical, Communications, and Cyber Systems Award ECCS-2221822. The data analysis was partially supported by the NSF through the Center for Dynamics and Control of Materials, an NSF Materials Research Science and Engineering Center (MRSEC) under Cooperative Agreement DMR-1720595, and the Welch Foundation Grant F-1814. The device fabrication work was supported by DARPA Microsystems Technology Office (MTO) Near Zero Power RF and Sensor Operations (N-ZERO) and COmpact Front-end Filters at the ElEment-level (COFFEE) project programs.

## Author contributions
R.L. and K.L. conceived the project. J.K and R.L. fabricated the LOBAR devices and performed VNA measurements. D.L. and S.J. performed the TMIM imaging and data analysis. D.L., R.L., and K.L. drafted the manuscript with contributions from all authors. All authors have given approval for the final version of the paper.

## Competing interests
The authors declare no competing interests.
