## [Peer Review File · Nature Communications]

REVIEWER COMMENTS

Reviewer #1 (Remarks to the Author):

Key results

The study reports on the characterization of a LiNbO₃ thin-film lateral overtone bulk acoustic resonator (LOBAR) by transmission-mode microwave impedance microscopy (TMIM). It is demonstrated that the modality is able to visualize multimode acoustic properties of the resonator in the 5 GHz regime with sub- μm spatial resolution and in-plane displacement sensitivity of 10 fm/ $\sqrt{\text{Hz}}$. Furthermore, the study experimentally evidences that relevant properties of a LOBAR device like anchor leakage, spurious modes and acoustic energy storage can be measured using TMIM.

Clarity and context

The text is written clearly, and the context of the results is well established. Relevant previous work, especially regarding TMIM and LOBARs, is considered and appropriately referenced. However, a previous report [1] of the authors, which is very closely related to the presented study, should be referenced more prominent in conjunction with the investigation of LiNbO₃ resonators, e.g. in line 66.

Minor suggestions to improve clarity:

- With the scan area being provided in lines 113-115, the scan speed would be of interest.
- Caption of Fig. 4: If understood correctly, the left image of 4a corresponds to the 32nd overtone. However, the current caption reads "TMIM images near the 32nd overtone of the LOBAR device". This suggests that all three images represent higher-order modes.

Validity

The presented data are of high quality, and they are comprehensibly interpreted. Robust theory is given to an appropriate extent either in the study or in suitable references. The conclusions are mostly well supported by the data. Having said that, some claims of the study should be further elaborated as outlined in the following:

1. The presented maps of acoustic waves with wavelengths in the order of $1\mu\text{m}$ impressively illustrate sub- μm resolution of TMIM. However, the claim of sub-100nm spatial resolution in line 71 is neither explicitly deduced from data in the study nor backed-up by a respective reference.

2. To support the claim that the sensitivity can be improved by cryogenic measurements, two publications are cited (lines 170-173). While an improvement can certainly be expected, it is not clear how the stated value of 1 fm/√Hz is extracted from the publications. Please explain how you estimate this number.

3. Lines 163-165 state that FEA simulations of the piezoelectric potential resemble the corresponding experimental data at P=-68 dBm. This specific data cannot clearly be recognized in Fig. 5a. Adding the experimental line scan to Fig. 5c for comparison would support the claimed match between simulation and experiment.

4. Lines 152-153 describe that higher-order modes can be seen as small peaks in TMIM-Mod² data in Fig. 4c. The data point associated with the 3rd transverse mode, marked by an arrow, is at 4.991 GHz according to the axis scale. In Fig. 4a this transverse mode is mapped at 4.992 GHz. Accordingly, the frequency of Fig. 4a does not fit to the peak position in Fig. 4c (especially since there is a data point in Fig. 4c at 4.992 GHz beside the peak). Please comment on this apparent inconsistency.

Significance

The development of MEMS devices accessing the GHz regime is a significant field of current research and development, driven by demands of key technologies like communication and quantum computing. However, the necessary understanding for any technological advancement is based on the ability to precisely measure relevant quantities and features. Although the authors already demonstrated the sub-μm spatial resolution of TMIM and measured the wave propagation for acoustic waves in LiNbO₃ thin films [1] and SAW devices [2], this work further advances the TMIM in terms of frequency regime, sensitivity, and energy loss modes. With the demonstrated capability to characterize acoustic resonators in the GHz regime, the study promotes TMIM as a valuable tool for further development of such devices.

[1] Lee, D., Meyer, S., Gong, S., Lu, R. and Lai, K. Visualization of acoustic power flow in suspended thin-film lithium niobate phononic devices. *Appl. Phys. Lett.* 119, 214101 (2021).

[2] Zheng, L., Wu, D., Wu, X. and Lai, K. Visualization of surface-acoustic-wave potential by transmission-mode microwave impedance microscopy. *Phys. Rev. Appl.* 9, 061002 (2018).

Reviewer #2 (Remarks to the Author):

The present work reports the application of an imaging system based on transmission-mode microwave impedance microscopy (TMIM). The method is applied for characterising a bulk acoustic resonator, which is known to operate at high frequencies and with a high quality factors. The parameters extracted with the present setup are the standard ones for resonators as e.g., resonance frequencies of the modes and their associated quality factor, and by comparing the results of the present investigation with finite element simulations the authors claim an in-plane sensitivity of $10 \text{ fm}/\sqrt{\text{Hz}}$.

A general remark on the work is necessary. The quality of the reported results is indeed high and the used experimental facilities are remarkable. Nevertheless, the degree innovation brought by the present work is not appealing. As claimed by the authors in the introduction, the TMIM was extensively treated by the same authors in past works (ref.s 25 and 26), and the characterised resonator type (LOBAR) appears in scientific papers since five years (one of the first paper on the topic is dated to 2018). The only relevant novelty of the article is the sensitivity of the experimental setup, which is validated with finite element simulations not properly detailed in the work. Therefore, I would suggest the possibility to move this work to another journal that is more focused on the improvement/application of known methods.

Some considerations that would improve the quality of the manuscript:

- The sensitivity of the imaging system is really high and it seems the major claim of the work. Is this achievement the result of some changes in the measurement setup compared to past works published by the same authors (ref. 25 and 26)?
- The authors declare that they used COMSOL to simulate the device and that they used material properties from COMSOL material libraries. How can they ensure that their device has said properties? COMSOL libraries are usually for bulk materials or films produced in very specific conditions, hence they are rarely applicable to systems with nanometric features.
- Regarding the adopted numerical model, some details on the solved equations would be appreciated and it would better clarify the claimed performance of the experimental setup.
- The authors use the term “spurious modes”. This term, especially when applied to MEMS resonators, identifies coupled modes that lead to nonlinearities in the dynamic response of the driven mode. The literature on the topic is vast. Regarding experiments in this setting I suggest the works by Steven W. Shaw or Mohammad Younis. Since the authors claim that the characterisation of spurious modes is necessary it would be nice to observe some analysis.

Point-by-point Response to Reviewers' Comments

Reply to Reviewer 1's report:

Reviewer Comment #1-1: *Key Result: The study reports on the characterization of a LiNbO₃ thin-film lateral overtone bulk acoustic resonator (LOBAR) by transmission-mode microwave impedance microscopy (TMIM). It is demonstrated that the modality is able to visualize multimode acoustic properties of the resonator in the 5 GHz regime with sub- μm spatial resolution and in-plane displacement sensitivity of $10\text{ fm}/\sqrt{\text{Hz}}$. Furthermore, the study experimentally evidences that relevant properties of a LOBAR device like anchor leakage, spurious modes and acoustic energy storage can be measured using TMIM.*

Clarity and context: The text is written clearly, and the context of the results is well established. Relevant previous work, especially regarding TMIM and LOBARs, is considered and appropriately referenced. However, a previous report [1] of the authors, which is very closely related to the presented study, should be referenced more prominent in conjunction with the investigation of LiNbO₃ resonators, e.g. in line 66.

Reply #1-1: We thank the reviewer for the excellent summary of key results in our work. While the mentioned paper was already cited as Ref. [33] in the previous submission, we agree with him/her that it should be referenced more prominently as early as in Line 66 or Ref. [27] rather than appearing later.

Changes #1-1: The previous Ref. [33] is now moved up to Ref. [27]. Line 66 now reads as "... transmission-mode microwave impedance microscopy (TMIM).^{25,26,27}" Labeling of references is adjusted accordingly.

Reviewer Comment #1-2: *Minor suggestions to improve clarity:*

- *With the scan area being provided in lines 113-115, the scan speed would be of interest.*
- *Caption of Fig. 4: If understood correctly, the left image of 4a corresponds to the 32nd overtone. However, the current caption reads "TMIM images near the 32nd overtone of the LOBAR device". This suggests that all three images represent higher-order modes.*

Reply #1-2: We thank the reviewer for carefully proofreading our manuscript. The scan speed of TMIM in this work is 3.33 sec per line (half time in the forward scan and the other half in backward scan). For Fig. 4a, all three images were taken around the 32nd overtone of the LOBAR device. The leftmost image corresponds to the fundamental mode of this overtone. The middle and rightmost images, with one and two nodes in the transverse direction, represent the 1st and 2nd order spurious modes of the same overtone, respectively. We have revised the caption of Fig. 4 for better clarity.

Changes #1-2: Lines 115 – 116 is now revised as "... are displayed in Fig. 2b. The scan speed is 3.33 sec per line (half time in the forward scan and the other half in backward scan) ...".

The caption of Fig. 4a is modified as "Left to right: TMIM images corresponding to the fundamental, 2nd, and 3rd transverse modes of the 32nd overtone of the LOBAR device..."

Review Comment #1-3-1: Validity: *The presented data are of high quality, and they are comprehensibly interpreted. Robust theory is given to an appropriate extent either in the study or in suitable references. The conclusions are mostly well supported by the data. Having said that, some claims of the study should be further elaborated as outlined in the following:*

1. *The presented maps of acoustic waves with wavelengths in the order of $1\ \mu\text{m}$ impressively illustrate sub- μm resolution of TMIM. However, the claim of sub-100nm spatial resolution in line 71 is neither explicitly deduced from data in the study nor backed-up by a respective reference.*

Reply #1-3-1: We are grateful for this critical comment from the reviewer. Indeed, while the spatial resolution of TMIM is presumably determined by the tip apex ($\sim 100\ \text{nm}$), we have not explicitly deduced such a number from experimental data. In this work, for instance, the acoustic wavelength on suspended LiNbO_3 thin films is $2 \sim 3\ \mu\text{m}$ at 5 GHz (phase velocity $\sim 12\ \text{km/s}$), as pointed out by the reviewer. Interestingly, on the unsuspended region, the phase velocity drops to $\sim 4\ \text{km/s}$ due to the substrate clamping and the corresponding wavelength is below $1\ \mu\text{m}$. As shown in Fig. R1, the TMIM can easily resolve half of a wavelength $\sim 0.4\ \mu\text{m}$. With this new data, we believe it is reasonable to claim a spatial resolution on the order of 100 nm.

Fig. R1 (Fig. S2) | Small features resolved by the TMIM. a TMIM image near the electrode. The bottom region is unsuspended and clamped by the Si substrate (see Fig. 1a in the main text). **b** Schematic of the LOBAR device with the region in (a) marked by the dash-dotted rectangle. The inset in the bottom is a close-up view of the red box in (a). The scale bar is $1\ \mu\text{m}$. **c** TMIM signal across the dashed line in (b). The acoustic wave with a wavelength of $\sim 0.8\ \mu\text{m}$ is clearly resolved.

Changes #1-3-1: In Line 71 of the main text, we changed the claim to “...individual tones with a spatial resolution on the order of 100 nm, from which...”. In Lines 128 – 130, we include a new sentence “In addition, the TMIM data in the unsuspended region of the device indicates that the spatial resolution is indeed on the order of 100 nm (Supplementary Information S2).” Moreover, Fig. R1 and the caption are added to Supplementary Information as Fig. S2. We also include a paragraph to describe the spatial resolution in Supplementary Information S2 – “Fig. S2a shows the TMIM image near an electrode of the LOBAR device. The bottom region of the LiNbO_3 thin film is unsuspended and clamped by the Si substrate. The phase velocity drops from $\sim 12\ \text{km/s}$ on the freestanding film to $\sim 4\ \text{km/s}$ on the unreleased film, resulting in a wavelength of $\sim 0.8\ \mu\text{m}$ here. The close-up image in Fig. S2b and the line profile in Fig. S2c both indicate that the TMIM can easily resolve lateral features on the order of 100 nm.”

Review Comment #1-3-2: 2. To support the claim that the sensitivity can be improved by cryogenic measurements, two publications are cited (lines 170-173). While an improvement can certainly be expected, it is not clear how the stated value of $1 \text{ fm}/\sqrt{\text{Hz}}$ is extracted from the publications. Please explain how you estimate this number.

Reply #1-3-2: The reviewer raised a very important issue: the TMIM sensitivity discussion was too brief in the previous submission. In order to quantitatively evaluate the sensitivity of our setup, we have carried out the standard noise power analysis in the revised manuscript. Fig. R2a shows the power spectral density (PSD) of several TMIM line scans. The acoustic signals in this LOBAR device correspond to $7 - 10 \text{ Hz}$ peaks in the Fourier transformation spectrum since they are oscillatory in space and thus oscillatory in time during line scans. The PSD peaks at $\sim 55 \text{ Hz}$ and higher harmonics are due to the power-line noise (the small difference between 55 Hz and 60 Hz may come from the data taking time). Using the conversion factor of $\sim 0.01 \text{ mV}/\text{fm}$ (see Reply #1-3-3 below) and taking the square root of PSD, we obtain the amplitude spectral density (APD) curves in Fig. R2b, from which a number of key features can be identified. First, the TMIM noise floor is limited by the power-line noise, around $10 \text{ fm}/\sqrt{\text{Hz}}$ in our instrument. Secondly, at $P_{\text{in}} = -68 \text{ dBm}$, the acoustic-wave signal is just above the power-line noise, consistent with the TMIM data at this input. Finally, the broadband noise is at a level of $3 \sim 5 \text{ fm}/\sqrt{\text{Hz}}$, which determines the ultimate sensitivity floor of our setup. In other words, even after filtering out the characteristic noise peaks, the acoustic signals cannot be observed for P_{in} below $-74 \sim -77 \text{ dBm}$. Further improvement of the sensitivity requires lowering such white noise, presumably the thermal Johnson-Nyquist noise. Since the thermal noise amplitude scales with the square root of measurement temperature, the ASD of the white noise will drop from $3 \sim 5 \text{ fm}/\sqrt{\text{Hz}}$ at room temperature (300 K) to below $1 \text{ fm}/\sqrt{\text{Hz}}$ under liquid-helium temperatures ($\sim 4 \text{ K}$) and below $0.1 \text{ fm}/\sqrt{\text{Hz}}$ at dilution-refrigerator temperatures ($< 100 \text{ mK}$). The analysis above explains how we estimate the sensitivity floor of the TMIM technique both in our setup at present and under cryogenic environments in the future.

Fig. R2 (Fig. 6) | Noise power analysis of TMIM data. a Power spectral density (PSD) of selected TMIM line scans with different input power P_{in} . **b** Amplitude spectral density (ASD) of the same data in (a). Peaks associated with the acoustic signals in this LOBAR device and power-line noise are labeled in both plots.

Change #1-3-2: In the revised manuscript, we have completely rewritten the discussion of the sensitivity floor in a new paragraph below and added a new figure Fig. 6 (see Fig. R2 above).

“In order to quantitatively evaluate the sensitivity of our instrument, we have carried out the standard noise power analysis⁴⁰ of the TMIM data. Fig. 6a shows the power spectral density (PSD) curves of selected TMIM line scans. The acoustic signals in this LOBAR device are easily identified as peaks at 7 – 10 Hz in the Fourier transformation spectrum since they are oscillatory in space and thus oscillatory in time during line scans. The PSD peaks at ~ 55 Hz and higher harmonics correspond to the power-line noise. Using the conversion factor above and taking the square root of PSD, we obtain the amplitude spectral density (APD) curves in Fig. 6b, from which a number of key features can be identified. First, the noise floor of our current setup is limited by the power-line noise of 10 fm/√Hz, which is already superior to the best value obtained by optical methods by more than a factor of 5.^{21,24} Secondly, the APD of the acoustic signal at $P_{in} = -68$ dBm is just above that of the power-line noise, consistent with the plot in Fig. 5b. Finally, the level of broadband noise in Fig. 6b is 3 ~ 5 fm/√Hz, which sets the ultimate sensitivity floor of our instrument. In other words, the acoustic signals cannot be observed for P_{in} below $-74 \sim -77$ dBm even after filtering out the power-line noise. Further improvement of the sensitivity requires the lowering of such Johnson-Nyquist white noise.^{41,42} Since the thermal noise amplitude scales with the square root of temperature, the ASD of white noise will drop from 3 ~ 5 fm/√Hz at room temperature (300 K) to below 1 fm/√Hz under liquid-helium temperatures (~ 4 K) and below 0.1 fm/√Hz at dilution-refrigerator temperatures (< 100 mK). Once the TMIM is implemented under cryogenic environments,^{43,44} the technique with unprecedented sensitivity will find widespread applications in the burgeoning field of quantum acoustics.”

We also added one reference Ref. [40] on the noise power analysis and two Refs. [41,42] on Johnson-Nyquist noise in the revised manuscript.

[40] Stoica, P. and Moses, R. *Spectral Analysis of Signals*, Prentice Hall, (2005).

[41] Johnson, J. Thermal Agitation of Electricity in Conductors. *Phys. Rev.* **32**, 97 (1928).

[42] Nyquist, H. Thermal Agitation of Electric Charge in Conductors. *Phys. Rev.* **32**, 110 (1928).

Review Comment #1-3-3: 3. Lines 163-165 state that FEA simulations of the piezoelectric potential resemble the corresponding experimental data at $P=-68$ dBm. This specific data cannot clearly be recognized in Fig. 5a. Adding the experimental line scan to Fig. 5c for comparison would support the claimed match between simulation and experiment.

Reply #1-3-3: The reviewer is correct that the specific data at $P_{in} = -68$ dBm cannot be recognized in Fig. 5a because this figure shows the line scans at all input powers. Since the acoustic signal at $P_{in} = -68$ dBm is just above the noise level, a direct comparison between simulation and experiment at this P_{in} is not compelling. Alternatively, we justify the claim as follows. Fig. R3a shows the TMIM signals at low input powers. As marked by the dash-dotted lines, the peaks near the electrodes are well resolved at $P_{in} = -65$ dBm, just resolved at -68 dBm, and barely discernible at -71 dBm, which supports our claim of the noise floor. In Fig. R3b, we plot the TMIM data, simulated piezoelectric potential, and simulated in-plane displacement at a higher $P_{in} = -53$ dBm, where the acoustic signals are much more prominent. The resemblance between the FEA results and TMIM line-scan data supports the claimed match between simulation and experiment. Finally, by comparing the peak TMIM signals and the simulated amplitude of in-plane displacements at all input powers in Fig. R3c, we obtain the conversion factor of ~ 0.01 mV/fm in our current setup.

Fig. R3 (Fig. S5) | Comparison between TMIM signals and FEA simulation. **a** TMIM line profiles at small or no input power. The dash-dotted lines mark the peak positions near the electrodes. **b** Top to bottom: Measured TMIM signals, simulated piezoelectric potential, and simulated in-plane displacement at a higher $P_{in} = -53$ dBm. **c** Peak TMIM signals versus simulated oscillation amplitude at all input powers. The red dashed line is a linear fit to the data with a slope ~ 0.01 mV/fm.

Changes #1-3-3: In the revised manuscript, we add two sentences to the main text as follows.

Lines 167 – 168: “A few TMIM line-scan data around this input power can be found in Supplementary Information S5.”

Lines 172 – 175: “Finally, by comparing the peak TMIM signals and the simulated amplitude of in-plane displacements at all input powers, we obtain the conversion factor of ~ 0.01 mV/fm in our current setup, as detailed in Supplementary Information S5.”

Fig. R3 and its caption above are now included as Fig. S5 in the Supplementary Information. The description of this supplementary figure is as follows. “Fig. S5a shows the TMIM signals at low input powers near the noise floor. As marked by the dash-dotted lines, the peaks near electrodes are well resolved at $P_{in} = -65$ dBm, just resolved at -68 dBm, and barely discernible at -71 dBm, which are consistent with Fig. 5b in the main text. Note that since the acoustic signal at $P_{in} = -68$ dBm is just above the noise level, a direct comparison between simulation and experiment here is not compelling. In Fig. S5b, we plot the TMIM data, simulated piezoelectric potential, and simulated in-plane displacement at a higher $P_{in} = -53$ dBm, where the acoustic signals are much more prominent. The resemblance between the FEA results and TMIM data supports the claimed match between simulation and experiment. Finally, by comparing the peak TMIM signals and the simulated amplitude of in-plane displacements at all input powers, we obtain the conversion factor of ~ 0.01 mV/fm in our current setup, as seen in Fig. S5c.”

Review Comment #1-3-4: 4. Lines 152-153 describe that higher-order modes can be seen as small peaks in TMIM-Mod² data in Fig. 4c. The data point associated with the 3rd transverse mode, marked by an arrow, is at 4.991 GHz according to the axis scale. In Fig. 4a this transverse mode is mapped at 4.992 GHz. Accordingly, the frequency of Fig. 4a does not fit to the peak position in Fig. 4c (especially since there is a data point in Fig. 4c at 4.992 GHz beside the peak). Please comment on this apparent inconsistency.

Reply #1-3-4: We thank the reviewer for carefully inspecting the figures. An omission caused the inconsistency in the previous version, which can be resolved by clearly labeling the frequencies. The three TMIM images in Fig. 4a were taken at 4.984, 4.987, and 4.991 GHz, respectively. In Fig. 4b, we presented three FEA simulated potential maps at 4.984, 4.988, and 4.992 GHz, which best match the experimental images. The slight difference in frequency between the experiment and simulation is likely due to uncertainties in device parameters (length, width, thickness, etc.).

Changes #1-3-4: In the revised Fig. 4, we carefully label the frequencies of the TMIM images in Fig. 4a and simulated patterns in Fig. 4b. The caption of Fig. 4 is also modified for clarity. The changes are summarized below as Fig. R4 (Fig. 4 in the main text).

Fig. R4 (Fig. 4) | Higher-order transverse modes and Q -factor. **a** Left to right: TMIM images corresponding to the fundamental, 2nd, and 3rd transverse modes of the 32nd overtone of the LOBAR device. **b** Simulated potential maps that match the experimental data. The slight difference in frequencies between (a) and (b) is likely due to uncertainties in device parameters. ...

Review Comment #1-4: Significance

The development of MEMS devices accessing the GHz regime is a significant field of current research and development, driven by demands of key technologies like communication and quantum computing. However, the necessary understanding for any technological advancement is based on the ability to precisely measure relevant quantities and features. Although the authors already demonstrated the sub- μm spatial resolution of TMIM and measured the wave propagation for acoustic waves in LiNbO₃ thin films [1] and SAW devices [2], this work further advances the TMIM in terms of frequency regime, sensitivity, and energy loss modes. With the demonstrated capability to characterize acoustic resonators in the GHz regime, the study promotes TMIM as a valuable tool for further development of such devices.

Reply #1-4: We are very grateful to the reviewer for his/her compliments on the significance of our work and the contribution to MEMS research. With all his/her comments addressed in the revised manuscript, we believe the paper is appropriate for publication in *Nature Communications*.

Reply to Reviewer 2's report:

Review Comment #2-1: *The present work reports the application of an imaging system based on transmission-mode microwave impedance microscopy (TMIM). The method is applied for characterising a bulk acoustic resonator, which is known to operate at high frequencies and with a high quality factors. The parameters extracted with the present setup are the standard ones for resonators as e.g., resonance frequencies of the modes and their associated quality factor, and by comparing the results of the present investigation with finite element simulations the authors claim an in-plane sensitivity of $10 \text{ fm}/\sqrt{\text{Hz}}$.*

A general remark on the work is necessary. The quality of the reported results is indeed high and the used experimental facilities are remarkable. Nevertheless, the degree innovation brought by the present work is not appealing. As claimed by the authors in the introduction, the TMIM was extensively treated by the same authors in past works (ref.s 25 and 26), and the characterised resonator type (LOBAR) appears in scientific papers since five years (one of the first paper on the topic is dated to 2018). The only relevant novelty of the article is the sensitivity of the experimental setup, which is validated with finite element simulations not properly detailed in the work. Therefore, I would suggest the possibility to move this work to another journal that is more focused on the improvement/application of known methods.

Reply #2-1: We thank the reviewer for his/her compliments that “*The quality of the reported results is indeed high and the used experimental facilities are remarkable*”. On the other hand, we respectfully disagree with him/her that “*the degree innovation brought by the present work is not appealing*”. Spatial imaging of MEMS devices provides crucial information that is not accessible through traditional vector network analysis. Research in this field is routinely featured in high-impact journals such as *Nature Communications*, e.g., Ref. [24] in our manuscript. Most works to date, however, have been using optical interferometric methods with limited spatial resolution and displacement sensitivity. The innovations of this TMIM work include (1) the first time that an acoustic resonator operating in the regime relevant to 5G technology and quantum acoustics is imaged with 100-nm spatial resolution; (2) the first time that acoustic modes with *in-plane* displacement are visualized in the real space; (3) the first time that an unprecedented sensitivity of $10 \text{ fm}/\sqrt{\text{Hz}}$ is reported in acoustic wave imaging. In addition to extracting standard parameters such as “*resonance frequencies of the modes and their associated quality factor*”, the high resolution and sensitivity in this work allow us to directly evaluate the acoustic leakage associated with the anchor loss and the presence of high-order transverse modes near certain overtones. We would also like to quote Reviewer #1's comment that “*this work further advances the TMIM in terms of frequency regime, sensitivity, and energy loss modes... to characterize acoustic resonators in the GHz regime*”. There is no doubt that our work contains sufficient novelty that warrants publication in *Nature Communications*.

Review Comment #2-2: *Some considerations that would improve the quality of the manuscript: The sensitivity of the imaging system is really high, and it seems the major claim of the work. Is this achievement the result of some changes in the measurement setup compared to past works published by the same authors (ref. 25 and 26)?*

Reply #2-2: We are glad that the reviewer appreciates the very high sensitivity of our imaging system, which is indeed one of the major advantages of this work. Compared with our past works (Refs. 25 and 26) with an operation frequency of ~ 1 GHz, the TMIM electronics in this work can cover the frequency band in the 4 \sim 6 GHz regime. Note that this major advancement in the operation frequency has already been highlighted in the manuscript. With that said, we understand that the sensitivity of our setup has never been quantitatively analyzed in previous publications, which may have caused the confusion. In the revised manuscript, we completely rewrote the discussion of the sensitivity floor by using the standard noise power analysis. Specifically, we performed fast Fourier transformation of the TMIM line scans and obtained the power spectral density (PSD) curves, from which we could evaluate the signal strength, power-line noise, and broadband (white) noise. The results quantitatively show that the sensitivity of our current setup is indeed $10 \text{ fm}/\sqrt{\text{Hz}}$, which can be further improved by reducing the power-line noise and cooling down the system.

Change #2-2: We add a new figure (Fig. 6) and a new paragraph (Lines 176 – 195) to the revised manuscript to describe the noise power analysis. The changes are reproduced as follows.

Fig. 6 | Noise power analysis of TMIM data. a Power spectral density (PSD) of selected TMIM line scans with different input power P_{in} . **b** Amplitude spectral density (ASD) of the same data in (a). Peaks associated with the acoustic signals in this LOBAR device and power-line noise are labeled in both plots.

“In order to quantitatively evaluate the sensitivity of our instrument, we have carried out the standard noise power analysis⁴⁰ of the TMIM data. Fig. 6a shows the power spectral density (PSD) curves of selected TMIM line scans. The acoustic signals in this LOBAR device are easily identified as peaks at 7 – 10 Hz in the Fourier transformation spectrum since they are oscillatory in space and thus oscillatory in time during line scans. The PSD peaks at ~ 55 Hz and higher harmonics correspond to the power-line noise. Using the conversion factor above and taking the square root of PSD, we obtain the amplitude spectral density (APD) curves in Fig. 6b, from which a number of key features can be identified. First, the noise floor of our current setup is limited by the power-line noise of $10 \text{ fm}/\sqrt{\text{Hz}}$, which is already superior to the best value obtained by optical methods by more than a factor of 5.^{21,24} Secondly, the APD of the acoustic signal at $P_{\text{in}} = -68$ dBm is just above that of the power-line noise, consistent with the plot in Fig. 5b. Finally, the level of broadband noise in Fig. 6b is $3 \sim 5 \text{ fm}/\sqrt{\text{Hz}}$, which sets the ultimate sensitivity floor of our instrument. In other words, the acoustic signals cannot be observed for P_{in} below $-74 \sim -77$ dBm

even after filtering out the power-line noise. Further improvement of the sensitivity requires the lowering of such Johnson-Nyquist white noise.^{41,42} Since the thermal noise amplitude scales with the square root of temperature, the ASD of white noise will drop from 3 ~ 5 fm/ $\sqrt{\text{Hz}}$ at room temperature (300 K) to below 1 fm/ $\sqrt{\text{Hz}}$ under liquid-helium temperatures (~ 4 K) and below 0.1 fm/ $\sqrt{\text{Hz}}$ at dilution-refrigerator temperatures (< 100 mK). Once the TMIM is implemented under cryogenic environments,^{43,44} the technique with unprecedented sensitivity will find widespread applications in the burgeoning field of quantum acoustics.”

Review Comment #2-3: *The authors declare that they used COMSOL to simulate the device and that they used material properties from COMSOL material libraries. How can they ensure that their device has said properties? COMSOL libraries are usually for bulk materials or films produced in very specific conditions, hence they are rarely applicable to systems with nanometric features.*

Reply #2-3: We thank the reviewer for raising this important point. For sputtered piezoelectric films, the deposition method and film thicknesses could strongly affect the quality, which may significantly deviate from bulk properties used in COMSOL libraries. On the other hand, we are using LiNbO₃ films that are thinned down from single-crystal wafers and transferred to Si substrates in this work. It has been reported that transferred LiNbO₃ films retain high crystallinity and bulk material properties even down to 100 nm (see Bousquet et al. 2020 IEEE International Ultrasonics Symposium pp. 1-4). As a result, it is a common practice to use the Piezoelectric Effect Multiphysics module and material libraries in COMSOL to simulate acoustic thin-film resonators. Examples include Refs. 9, 10, 28 – 32 in our manuscript, where COMSOL simulations are in good agreement with experimental results measured by vector network analyzers (VNAs). In the inset of Fig. 1b in our manuscript, we also show that the measured admittance response of our device matches very well with results from the simulation, which provides compelling evidence on the effectiveness of the FEA modeling. In all, we believe that the use of COMSOL material libraries for our LOBAR device with transferred LiNbO₃ thin films is well justified. A few changes to the manuscript are listed below for clarification.

Change #2-3: In Lines 91 – 92 of the revised manuscript, we add a new sentence to comment on the FEA result – “The excellent agreement between the VNA data and simulated curves highlights the effectiveness of FEA modeling.”

In the Methods / Device fabrication Section, we revise the first two sentences as “The 510 nm 128° Y-cut LiNbO₃ thin film, which was thinned down from a single-crystal wafer and transferred onto a 4-inch silicon (Si) wafer, was provided by NGK Insulators, Ltd.²⁹ Such films retain high crystallinity and bulk material properties.⁴⁵”

In the Methods / Finite-element modeling Section, we revise the first sentence (Lines 216 – 217) under as “...using COMSOL Multiphysics 6.0 software, which is widely used for the simulation of acoustic thin-film resonators.”

A new reference (Ref. [45]) is added to the bibliography.

45. Bousquet, M. et al. Lithium niobate film bulk acoustic wave resonator for sub-6 GHz filters. 2020 IEEE International Ultrasonics Symposium (IUS), Las Vegas, NV, USA, 2020, pp. 1-4.

Review Comment #2-4: *Regarding the adopted numerical model, some details on the solved equations would be appreciated and it would better clarify the claimed performance of the experimental setup.*

Reply #2-4: We thank the reviewer for this suggestion. According to the Piezoelectric Effect Multiphysics module in COMSOL, the software solves elastic wave equations in a linear piezoelectric medium with strain-charge coupling. As detailed in our Reply #2-3 above, the simulation agrees well with the electrical characterization results. On the other hand, we note that the emphasis of this work is the acoustic imaging and the performance of the TMIM setup has nothing to do with the numerical modeling.

Change #2-4: In the Methods / Finite-element modeling Section, we add a sentence (Lines 219 – 220) as “The software solves elastic wave equations in a linear piezoelectric medium with strain-charge coupling.”

Review Comment #2-5: *The authors use the term “spurious modes”. This term, especially when applied to MEMS resonators, identifies coupled modes that lead to nonlinearities in the dynamic response of the driven mode. The literature on the topic is vast. Regarding experiments in this setting I suggest the works by Steven W. Shaw or Mohammad Younis. Since the authors claim that the characterisation of spurious modes is necessary it would be nice to observe some analysis.*

Reply #2-5: We thank the reviewer for pointing out the literature on nonlinear dynamic response of the driven mode. In our work, we follow the definition of spurious modes in GHz MEMS resonators (see Ref. [1]), which refer to undesired acoustic resonances near the high-Q main tones. Specifically, the spatial patterns in Fig. 4a with zero, one, and two nodes correspond to the fundamental, 2nd, and 3rd transverse standing waves in the LOBAR device, respectively. These higher-order resonant modes have nothing to do with nonlinearities under dynamic driving. In order to avoid confusions, we modify the text as follows.

Changes #2-5: We add one sentence in Lines 150 – 152 to clarify that the term ‘spurious mode’ in this paper “Note that these undesired spurious resonances are different from the ones due to nonlinear dynamic response of the drive mode.^{38,39}”. We also cite two papers by S.W. Shaw and M. Younis on the nonlinear dynamic response of the drive mode.

38. Ruzziconi, L., Jaber, N., Kosuru, L., Bellaredj, M.L. and Younis, M.I. Internal resonance in the higher-order modes of a MEMS beam: experiments and global analysis. *Nonlinear Dyn.* **103**, 2197 (2021).

39. Shoshani, O. and Shaw, S.W. Resonant modal interactions in micro/nano-mechanical structures. *Nonlinear Dyn.* **104**, 1801 (2021).

In summary, we have answered all questions from the reviewer and addressed his/her comments in the revised manuscript. We sincerely hope that he/she will agree that our paper is now ready for publication in *Nature Communications*.

REVIEWERS' COMMENTS

Reviewer #1 (Remarks to the Author):

The authors have addressed all questions and comments from the initial review in a comprehensible and thorough manner. With these clarifications and changes, the article is ready for publication.

Reviewer #2 (Remarks to the Author):

Dear Authors, Dear Editor,

I am writing to you in regards to the article “Nanoscale Imaging of Super-High-Frequency Microelectromechanical Resonators with Femtometer Sensitivity”, which I have reviewed for publication in Nature Communications.

After reading and evaluating the revised manuscript, I am pleased to recommend it for publication. The authors have effectively addressed major and minor comments. Furthermore, they also improved the presentation of the novel aspects of their work.

Additionally, contrary to the opinion I had on the original submission, I believe the article now fits well within the scope and focus of Nature Communications. The methodology is sound and the results are robust.

In conclusion, I suggest the publication of the article in Nature Communications.

Reply to Reviewer's report:

Reviewer 1' Comment:

The authors have addressed all questions and comments from the initial review in a comprehensible and thorough manner. With these clarifications and changes, the article is ready for publication.

Reviewer 2' Comment:

Dear Authors, Dear Editor,

I am writing to you in regards to the article "Nanoscale Imaging of Super-High-Frequency Microelectromechanical Resonators with Femtometer Sensitivity", which I have reviewed for publication in Nature Communications.

After reading and evaluating the revised manuscript, I am pleased to recommend it for publication. The authors have effectively addressed major and minor comments. Furthermore, they also improved the presentation of the novel aspects of their work.

Additionally, contrary to the opinion I had on the original submission, I believe the article now fits well within the scope and focus of Nature Communications. The methodology is sound and the results are robust.

In conclusion, I suggest the publication of the article in Nature Communications.

Our responses : We are very grateful to receive constructive reviews from the referees. With his/her final comment addressed, we hope that the manuscript is now suitable for publication in *Nature Communications*. Since all the referees recommended publishing our work as it is, we did not change the manuscript this time except for some minor formatting.

Thank you very much for your consideration.

Best regards,

Keji Lai
Associate Professor of Physics, University of Texas at Austin
Austin, TX 78712 USA
Phone: (512) 475-9128
Email: kejilai@physics.utexas.edu